# The Role of the Innate Immune Response in Oral Mucositis Pathogenesis

**DOI:** 10.3390/ijms242216314

**Published:** 2023-11-14

**Authors:** Joanne Bowen, Courtney Cross

**Affiliations:** School of Biomedicine, University of Adelaide, Adelaide 5005, Australia; courtney.subramaniam@adelaide.edu.au

**Keywords:** oral mucositis, innate immunity, inflammation, interventions

## Abstract

Oral mucositis (OM) is a significant complication of cancer therapy with limited management strategies. Whilst inflammation is a central feature of destructive and ultimately ulcerative pathology, to date, attempts to mitigate damage via this mechanism have proven limited. A relatively underexamined aspect of OM development is the contribution of elements of the innate immune system. In particular, the role played by barriers, pattern recognition systems, and microbial composition in early damage signaling requires further investigation. As such, this review highlights the innate immune response as a potential focus for research to better understand OM pathogenesis and development of interventions for patients treated with radiotherapy and chemotherapy. Future areas of evaluation include manipulation of microbial–mucosal interactions to alter cytotoxic sensitivity, use of germ-free models, and translation of innate immune-targeted agents interrogated for mucosal injury in other regions of the alimentary canal into OM-based clinical trials.

## 1. Introduction

Mucositis is defined by the National Library of Medicine as inflammation of the mucosa caused by radiotherapy and chemotherapy. It has been traditionally studied as separate entities in different areas of the alimentary tract, namely, the oropharyngeal, intestinal, and rectal segments; however, there are likely many commonalities and overlapping features in mucositis pathogenesis regardless of region [1]. This review focuses on the mucosal changes visualized in the mouth and oropharynx, referred to as oral mucositis (OM), which is typified by self-limiting erythema, edema, mucosal ulceration, and pseudomembrane formation [2]. In OM, the progression from inflammation to ulcerative mucosal destruction and eventual repair follows a relatively well-defined trajectory, contemporarily described around two decades ago [3]. Some of the key mediators of OM include reactive oxygen species (ROS); DNA damage responses, including apoptosis; transcription factors, such as NF-κB; inflammatory cytokine cascades; submucosal signals that alter matrix deposition and support structures; and epithelial and vascular restitution signals. This is all overlayed with the impact of microbial-derived signals, including metabolites, alterations in redox status, and biofilm components, due to colonization at sites of ulceration [4].

Radiotherapy was the first cancer treatment modality to garner interest for the direct and indirect mucosal damaging properties of ionizing radiation affecting tissues adjacent to the tumor site [5]. OM has been reported to occur at least to some degree in up to 90% of patients receiving high dose radiotherapy for head and neck cancer [6]. Chemotherapy as a systemic treatment is associated with a myriad of toxicities affecting all body systems; however, the mucosal impacts are most frequently associated with the drug classes fluoropyrimidines and topoisomerase inhibitors and are reported in the ranges of 10 to 40% [7]. Patients’ experience of oral mucositis is highly variable but can be associated with their treatment; genetic predisposition; socioeconomic determinants; and other risk factors, such as age and weight [8]. In general terms, visible OM onset occurs for radiotherapy-alone regimens after a cumulative dose of approximately 30 Gy [9]. OM arises around one to two weeks into cyclical chemotherapy regimens and is more severe when the modalities are combined [10]. In the case of high-dose chemotherapy used in conditioning regimens for hematological stem cell transplant, OM has been reported in the majority of patients and increases the risk of infection and mortality during profound neutropenia [11]. 

The estimated costs associated with OM are hard to calculate based on worldwide differences in medical billing; however, research conducted in the US has suggested that OM-impacted regimens contribute over four billion US dollars annually [12] and cause incremental costs of at least USD 5000 per radiotherapy patient and USD 3700 per cycle of chemotherapy [13]. Increased expenditure occurs due to hospitalization and resource utilization to manage hydration, pain control, feeding tubes and parenteral nutrition, and infection control [14]. Mucosal toxicities are also experienced during targeted therapies, including mTORC1 and EGFR inhibitors, although they are typically reported as occurring at a lower incidence and severity [15]. The experience of oral injury with immunotherapies is now emerging as a toxicity to investigate, particularly given the relationship between immune checkpoint inhibitors and inflammation across systems [16]. However, since there is a lack of evidence in the area of targeted immunotherapies and OM, the following sections focus on conventional treatment modalities.

## 2. Current Management of OM

The approach to OM control during radiotherapy and chemotherapy has been extensively studied and summarized to generate clinical practice guidance [17]. The key practice guidance includes basic oral care (rinsing and hygiene) and pain control as well as photobiomodulation, anti-inflammatories (benzydamine), cryotherapy, and KGF-1 in certain settings [17]. Some practical limitations to establishing clear guidance is the array of OM measuring tools, making trial-to-trial comparisons challenging. Assessment tools also need to be updated to account for differences in presentation related to emerging treatment modalities and newer combination regimens as well as to have a strong patient perspective [18]. Regardless of these limitations, the approach to management of OM to date has not fully applied knowledge of injury and restorative processes. While symptoms such as pain rely on management through embedded pharmacological approaches [19], these do not target known mucosal characteristics of OM. Basic oral care with rinsing looks to deload microbial triggers of inflammation [20], while KGF-1 aims to increase recovery of mucosal integrity via stimulating proliferation of epithelium. Avasopasem manganese, a superoxide dismutase mimetic, targets upregulated ROS as an antioxidant enzyme; however, it has yet to be included in guidance documents despite showing benefit in reducing severe OM in recent clinical trials for H&N cancer treated with chemoradiotherapy [21]. 

A number of clinical studies have investigated anti-inflammatory drugs, including benzydamine, celecoxib, irsogladine maleate, misoprostol, and rebamipide, with only benzydamine mouthwash reaching evidence sufficient to warrant a practice guideline for prevention of OM [22]. This focus stems from the principle pathology associated with OM development, namely, inflammation that progresses to ulceration. While inflammatory processes are clearly a significant component of OM with increased proinflammatory cytokine concentrations consistently detected in clinical and preclinical models of OM [3], these changes may be a consequence of injury processes rather than the driving force. Given there have been limited degrees of success in this anti-inflammatory drug-mediated approach, targeting the end product, inflammation, appears to be an oversimplification and requires investigators to take a step back to reevaluate OM induction mechanisms and the contributing cellular responses. To this end, the innate immune response has become a hot topic as it allows investigators to combine epithelial, microbial, and immune interactions to evaluate development of injury.

## 3. The Innate Immune System in OM Development

The innate immune system comprises barriers, cells, and humoral factors, which are responsible for control of homeostatic tolerance to commensal microbes as well as indiscriminate host responses to potential physiological threats [23]. The constant threat of microbial breaches across the epithelium ensures that innate immunity is highly developed at the oral microbe–host interface [24]. The oral cavity has the second largest and diverse microbiota after the gut, harboring over 700 species of bacteria [25]. The presence of both hard and soft surfaces for colonization, contact with ingested material and airborne factors, and being bathed in saliva containing mixtures of peptides in circadian patterns means it is an incredibly complex environment [26]. A sophisticated network of interactions between the innate immune system and the oral microbiota creates a bidirectional relationship to maintain optimal physiology for both systems [27].

Although not classically considered a part of the innate immune system, epithelial cells provide physical exclusion of oral microbes and their products through the action of tight junctions [28]. Tight junctions are the main determinant of epithelial barrier function and are made up of transmembrane and intracellular proteins of the claudin, occludin, zonula occludens, and junctional adhesion molecule groups. Loss of tight junction proteins has been shown following both radiotherapy and chemotherapy and is considered important in mucositis development [29,30]. Overlaying the epithelium is mucus and antimicrobial peptides, including IgA and tree-foil factors secreted within saliva, providing an additional mucosal innate immune defense mechanism. Salivary gland function and saliva composition is rapidly altered in response to cancer therapies and impairs the ability to retain optimal conditions to balance microbial and epithelial interactions, including compositional shifts in the oral microbiome [31]. Epithelial cells lining the oral cavity express a range of innate immune sensors that mediate signals between microbes and immune cells [32] and have been shown to be major contributors of IL-1β secretion during oral inflammation [33]. IL-1β has attracted major attention in OM research due to its diverse role in injury development and inflammatory signals [34,35]. 

Where there is a loss of spatial separation between surface microbial factors and the lamina propria, the result is activation of oral mucosal innate immune cells proper, including macrophages, dendritic cells, neutrophils, and innate lymphoid cells. These cells have a range of microbial sensing systems designed to orchestrate a rapid response through the release of defense-associated factors, such as bacteriocins and cytokines [36]. In particular, type I macrophages have been associated with OM progression due to submucosal release of a range of proinflammatory cytokines and chemokines, including TNFα, IL-1β, IL-6, CXCL8, and MIP [37]. Infiltrating neutrophils also contribute to OM progression by enhancing the release of cytokines and MMPs and have been implicated in risk of severe OM development through an elevated neutrophil-to-lymphocyte ratio [38]. One intriguing factor is IL-17, which is released during an inflammatory response by innate immune cells and is associated with neutrophil recruitment [39]. In radiotherapy-induced OM, it is increased substantially in areas of ulceration and has been hypothesized to signal repair of tight junctions. In mice lacking the IL-17 receptor, OM severity is increased, presumably through inability to fine-tune neutrophil responses to restore epithelium [40] and lack of production of antimicrobial defensins [41]. Despite these intriguing findings, studies are yet to directly investigate IL-17 as an OM treatment. Finally, dendritic cells are commonly found at the oral epithelial barrier interface and responsible for integrating microbial signals to the immune system via pattern recognition receptors and antigen presentation [42].

### 3.1. Pattern Recognition Receptors and OM Development

Pattern recognition receptors (PRRs) in the innate immune system were first described over 30 years ago [43]. These receptors sense microorganisms through conserved molecular structures and comprise several families, including toll-like receptors (TLRs), nucleotide-binding oligomerization (NOD)-like receptors (NLRs), RIG-I-like receptors, C-type lectin receptors, absent in melanoma 2 (AIM2)-like receptors, and OAS-like receptors [44]. PRRs constitute a continuous surveillance system for the presence of pathogen-associated molecular patterns (PAMPs) and endogenous-damage-associated molecular patterns (DAMPs) and are expressed on both the external and internal cellular compartments of immune and epithelial cells [45]. 

TLRs have been increasingly recognized as being important in mucositis pathogenesis, established through a range of studies focusing predominantly on lower gastrointestinal tract injury. These studies clearly demonstrated modulation of mucosal injury following chemotherapy due to the knock-out of specific proteins, namely, TLR2 [46], TLR4 [47,48], and TLR9 [46]. However, whether genetic deletion proves protective or exacerbates injury depends on the specific model and context. Dependent on the external facing motif of each receptor, TLRs can recognize PAMPs, including lipoproteins and peptidoglycan, lipoteichoic acid (LTA), fungal zymosan, single and double-stranded RNA, lipopolysaccharide (LPS), bacterial flagellin, and double-stranded and unmethylated CpG DNA [49]. In addition, DAMPs, such as heat shock proteins; HMGB-1; and ECM molecules biglycan, tenascin-C, versican, hyaluronic acid, and heparan sulfate, are able to activate TLRs [50]. All TLRs are expressed in the oral mucosa and appear to be upregulated during periods of inflammation [51], establishing their potential role in OM pathology [52,53]. However, the spatiotemporal distribution and contribution of TLRs expressed on different cells within the oral cavity on the development of oral mucositis has yet to be described.

Whilst there is a lack of direct evidence of TLR signaling in controlling OM severity, TLRs clearly maintain homeostasis of the oral epithelium. For example, the TLR4 rs10759931 polymorphism is significantly associated with recurrent aphthous stomatitis, suggesting that TLR4 may be an important mediator of oral ulcers [54]. It is likely that TLR signaling in the oral cavity creates direct defensive mechanisms via the induction of antimicrobial substances, such as defensins, which influence microbial communities and function [55,56]. The downstream intracellular signaling pathways for all TLRs incorporate kinases and transcription factors either through the myeloid differentiation factor 88 (MyD88)-dependent pathway or the TIR domain containing adapter-inducing interferon β (TRIF)-dependent pathway [57]. Importantly, the transcription factor NF-κB is a key component activated by TLR signaling and is known to contribute to numerous inflammatory signals, including those leading to OM [58,59]. During ulcerative mucositis, PAMPs, such as lipopolysaccharides, can interact with TLRs expressed on innate immune cells of the oral mucosa to trigger an inflammatory response [60]. In addition, release of DAMPs, such as HMGB1, from cells undergoing necrosis or oxidative stress also contributes to innate immune activation via detection by TLRs. It has been demonstrated that HMGB1 overexpression exacerbates chemotherapy-induced oral inflammation and, furthermore, inhibition of HMGB1 production attenuates OM [61]. 

Whilst not as comprehensively studied as the TLR family, the NLRs are a class of pattern-recognition receptors that are also likely to play important roles in OM development [62]. These receptors are mostly expressed in the cytoplasm to detect intracellular danger signals [63], including bacterial cell products and cell stress markers, such as ROS [64]. Of key interest for OM is the formation of the NLRP3 inflammasome, which occurs after priming by lipopolysaccharide and mitochondrial dysfunction and leads to caspase-1 activation followed by IL-1 and IL-18 secretion [65]. Given the critical role of IL-1β in neutrophil recruitment and other consequences of OM, the inflammasome presents a compelling target for further research. 

### 3.2. Oral Microbiome—Innate Immune Interactions and OM Development

The oral microbiome is known to be altered by chemotherapy and radiotherapy, contributes to the severity of oral injury, and may be a predictive factor for risk of OM development [66]. The mechanisms by which the microbiome exerts this influence have yet to be fully uncovered; however, evidence supports a bidirectional relationship between these oral microbial communities and the innate immune system [67]. Structural and secretory changes in the mouth during treatment as well as the inflammatory processes underpinning OM development influence oral microbial communities. Typically a decrease in the numbers and diversity of microbial communities and a compositional shift from a commensal towards a pathobiont-dominated enterotype is observed following both chemotherapy and radiotherapy [68]. The loss of commensals undermines the normally highly regulated coordination between the oral microbiome and host innate immune system to maintain homeostasis [53]. The indirect impacts of altered host–microbial interactions can include dysfunctional signaling through immunological regulators, such as defensins, CXCL2, and GAS6 [69,70]. During ulcerative mucositis progression, increased bacterial colonization at these sites causes unbalanced PAMPs and DAMPs, which trigger PRR-mediated activation of downstream NF-κB signaling pathways, leading to augmented inflammatory responses due to the constant release of proinflammatory cytokines from innate immune cells [61]. The compositional changes to the oral microbiome observed across prior studies have varied, with reduction and expansion of differing microbial species being identified following treatment [71,72]. This varying nature of compositional analysis has led to suggestions that the resilience of the oral microbiome as well as the speed at which the microbial community is able to recover post-treatment may offer more reliable indicators for the impact of the oral microbiome on OM development and severity [73].

The introduction of germ-free mice in experimental research has elegantly confirmed the critical role of microbes in both OM development and shaping the innate immune system [74]. In a recent study, germ-free mice exhibited significantly less severe OM following treatment with the common chemotherapeutic 5-fluorouracil compared to specific-pathogen-free conventional mice. This was observed alongside a decrease in MMP and proinflammatory cytokine expression in the oral mucosa, underscoring the potentially causal role of the microbiome in OM development. However, given that germ-free mice are completely devoid of a microbiome, the results of this study cannot confirm the microbial community responsible, with both the oral and gut microbiome hypothesized to play a role in OM development in isolation to one another and collectively [75], likely through microbe–immune interactions. In relation to the innate immune system, myeloid cells are decreased in germ-free mice [76] and have reduced function, possibly explained by the diminished cytokine and chemokine levels necessary for cell recruitment and differentiation [77]. Additionally, innate-like lymphocytes and innate lymphoid cells that hone to barrier sites within the oral cavity are influenced by the presence of commensal microbes and mediate granulocyte activity [78]. Therefore, while research is still needed to precisely examine how microbiome–innate immunity crosstalk influences OM, there are clearly multiple points of interaction that could significantly impact injury processes via inflammation regulation (Figure 1). 

## 4. Innate Immune Response Targeted Interventions for OM

Undoubtedly, the application of new insights into the role of innate immune responses on OM development is to direct research towards novel effective interventions. And given that the innate immune system comprises barriers, cells, and humoral factors, all with known involvement in OM development, it provides a number of potential therapeutic targets for alleviating symptom burden. 

Since the appreciation of microbial disruption as a component of mucositis pathogenesis, multiple attempts have been made to sterilize the oral cavity during treatment with various antiseptics and antibiotics [79,80,81,82,83]. There has been a general lack of clinical effectiveness with this approach, suggesting oral microbial load is a minor contributor to OM severity. In comparison, the use of ingested probiotics has shown potential to reduce severe cases of OM following mixed cancer regimens [84] but not when applied as a lozenge [85]. Probiotics influence innate immunity via increasing B-cell-mediated IgA secretion and immunomodulation via dendritic cell cytokine release and TLR activation [86]. The risk with general depletion of microbes as a strategy to improve OM is the disruption to commensals that are required to maintain homeostasis and assist with mucosal repair processes through innate immune signaling. Alternatively, ingested probiotic-based therapies influence the gut microbiome and may indirectly enhance systemic immune responses to support oral integrity. This theory is encouraged by recent work that found mice treated with intragastric antibiotics had reduced severity of OM that occurred independent of changes in oral microbiome composition, indicating a systemic regulation of inflammation risk [87]. In additional experiments, the same investigators also found that oral microbiome transplantation from healthy mice was able to reduce OM severity, which corroborates the hypothesis that oral microbiome composition regulates oral mucositis [88], but further research is required to unpack the direction and causality.

Any shifts in microbes will undoubtedly change the composition of PAMPs; thus, interventions that directly target these molecules or their sensors (PRRs) will potentially be powerful regulators of innate immune responses in OM. The TLR5 agonist BLB502 (developed as CBLB502 entolimod) has been shown to be effective at alleviating mucositis following single-dose and fractionated radiotherapy to the head and neck [89]. Similarly, TLR5 agonist KMRC011 has been explored in multiple models of OM and found to provide some protection against ulcer development [90,91]. The proposed mechanisms of protection appear to be through NF-κB signaling, leading to superoxide dismutase 2 induction and granulocyte-colony-stimulating factor production. Topical melatonin has been shown to prevent NLRP3 inflammasome activation by reducing mitochondria oxidative damage and dysfunction [92]. Glycosaminoglycans, such as hyaluronic acid, are DAMPs known to activate TLRs. GM-1111 is a synthetic glycosaminoglycan molecule, which was found to reduce radiation-induced oral inflammation in mice by inhibiting TLR-mediated proinflammatory cell signaling and the NLRP3 inflammasome, highlighting the context-specific regulation of injury through these PRRs [93]. Dusquetide (SGX942) is a first-in-class innate defense regulator (IDR) that has shown benefit against radiotherapy-induced OM by modulating the innate immune response through a key adaptor protein known as p62 [94]. Golotimod (SCV-07), a synthetic peptide that acts broadly on the TLR pathway, has been shown to reduce injury in a hamster model of chemoradiotherapy-induced OM [95]. The interventions and their ability to modify oral mucositis progression have been summarized in Table 1.

## 5. Future Directions

Given the inherent similarities in pathophysiology, investigating the efficacy of drugs that have shown promise in the treatment of intestinal mucositis in the context of OM represents an important avenue for future research. Anakinra is a recombinant human IL-1 receptor antagonist, which has been shown to alleviate intestinal mucositis in various preclinical models [96,97,98]. Clinical Phase IIA trials of anakinra have also established safety in HSCT recipients, with Phase IIB trials currently establishing efficacy in the management of intestinal mucositis [98,99]. By blocking the action of IL-1β, anakinra controls the positive feedback loops through which IL-1β exacerbates inflammation to degrade the mucosal barrier lining. Considering this, along with the knowledge that IL-1β-induced epithelial barrier injury in the oral mucosa is a critical driver of OM development and progression, anakinra may hold promise in the treatment of OM. Neutrophil-targeted agents have received increasing recognition for their potential to treat mucositis, with a recent preclinical study establishing that MPH-996, a neutrophil elastase inhibitor that limits neutrophil degranulation and the resulting inflammation, is capable of reducing intestinal barrier damage, inflammation, and gut microbial imbalances [100]. With neutrophil-derived inflammation being an important hallmark of OM, MPH-996 may also hold potential in the treatment of OM, thus warranting further investigation.

The results of the growing body of work dedicated to exploring the efficacy of innate immune-targeted therapeutics in alleviating intestinal mucositis highlight a myriad of possible therapeutics for the treatment of OM. However, considering the highly comparable pathophysiological mechanisms underlying both intestinal and oral mucositis, analyzing both outcomes simultaneously in models treated with chemotherapy and radiotherapy represents an important consideration for future research to streamline the screening of potential therapeutics.

While the studies described represent a generally successful paradigm when harnessing innate immune responses to mitigate oral or intestinal inflammation, it is important to acknowledge the inherent limitations associated with such therapeutics, namely, their potential interactions within the tumor microenvironment and effects on immunogenic cell death. A number of anticancer therapies, including radiotherapy and certain chemotherapies, induce immunogenic cell death through promoting the release of DAMPs from dying tumor cells. Recognition of these DAMPs by innate PRRs, including TLRs, activates tumor-specific immune responses to enhance efficacy of anticancer agents through the combined action of direct cytotoxicity and antitumor immunity [101]. Given that the innate immune system is critical in establishing this response, therapeutics capable of influencing immune function could unwittingly impair or inhibit tumor response.

In addition to the potential effects on radiotherapy- and/or chemotherapy-related immunogenic cell death, OM therapies targeting the innate immune system could also influence the efficacy of immunotherapy. Increasingly, drugs that alter immune function are receiving recognition for their detrimental effects on immunotherapy outcomes. For example, in a cohort of patients with non-small-cell lung cancer receiving PD-(L)-1 blockade, corticosteroid use was associated with decreased overall response rate, progression-free survival, and overall survival [102]. Furthermore, antibiotics have also been associated with inferior clinical outcomes following immunotherapies [103,104], which is thought to be through alterations to the microbiome and the resulting implications for immune function. In the context of chemotherapy, mice that were treated with antibiotic or germ-free had reduced IL-17 responses and tumors resistant to cyclophosphamide [105]. In a similarly designed experiment, it was found that these mice failed to respond to oxaliplatin and had shorter survival, likely due to reduced tumor DNA damage and apoptosis, in part by decreased ROS production [106]. Collectively, these studies point to the interwoven relationship between mechanisms responsible for OM development and tumor cell cytotoxicity that needs to be considered during the development of any new intervention.

## 6. Conclusions

The innate immune response provides both protective and injury signals within the oral cavity depending on the context. These signals are balanced across epithelial, immune, and microbial compartments to regulate inflammation caused by radiotherapy and chemotherapy. The increasing appreciation of innate immunity and interactions with resident microbes during OM development has created opportunities to target specific features of early signaling cascades. This is an advance on traditional therapeutic interventions, which often aimed to intervene once inflammation was thoroughly established. Future directions for the field should include testing OM interventions in relevant tumor-bearing models to avoid inadvertent cancer cryoprotection and develop more complex models that take into account the effects of emerging immunotherapies and combination regimens on innate immune signals. Locally targeted strategies for OM prevention that do not negatively disrupt whole-gut microbial composition should be prioritized. As research continues to clarify the direct interactions between innate immune sensors and subsequent cellular processes, the payoff will be more precise and OM interventions will be more targeted, leading to improved outcomes for patients.

## Figures and Tables

**Figure 1 ijms-24-16314-f001:**
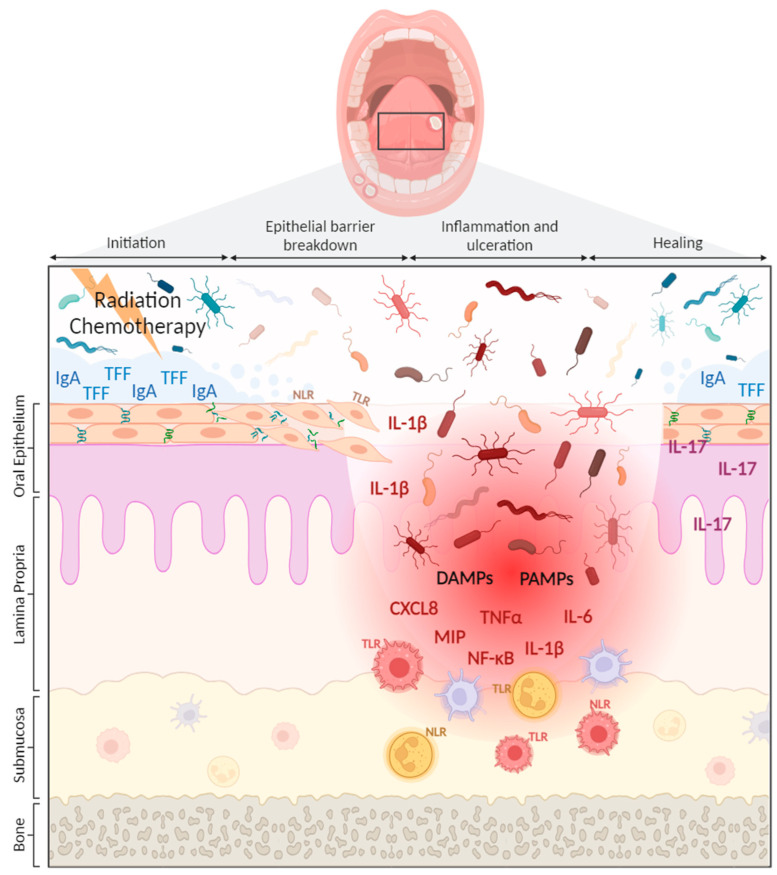
Role of the innate immune system in the development of OM. Initiation: Epithelial cells, connected via tight junctions, provide a physical barrier excluding oral microbes and their products from underlying tissue. Overlaying the epithelium is mucus and antimicrobial peptides, including IgA and tree-foil factors (TFF) secreted within saliva. At initiation of OM, the oral mucosa is exposed to radiotherapy or chemotherapy. Epithelial barrier breakdown: Following exposure, loss of tight junction proteins compromises epithelial barrier integrity. Salivary gland function and saliva composition is altered, decreasing its antimicrobial properties and influencing microbial and epithelial interactions. This induces compositional shifts in the oral microbiome, increasing the presence of pathobionts and enhancing IL-1β production by epithelial cells. Inflammation and ulceration: There is a loss of physical separation between surface microbes and the lamina propria, causing activation of oral mucosal innate immune cells, including macrophages, dendritic cells, neutrophils, and innate lymphoid cells, through interactions of PRRs (TLRs and NLRs) with PAMPs and DAMPs. OM progresses due to submucosal release of proinflammatory cytokines and chemokines, including TNFα, IL-1β, IL-6, CXCL8, and MIP. Healing: IL-17 plays an important role in fine-tuning neutrophil responses to trigger injury resolution and epithelium restoration.

**Table 1 ijms-24-16314-t001:** Innate immune response targeted interventions for oral mucositis.

Intervention Type		Research Methodology			Effective?		Reference
		Therapeutic Compound	Study Design	Study Subjects	Yes	No	
Clinical	Antiseptic/antibiotic approaches	Iodine-based mouthwash	Double-blind, randomized controlled trial (*n* = 20)	Patients receiving *chemoradiation* for head and neck cancer		X	[79]
		0.3% chlorhexidine-based mouthwash	Double-blind, randomized controlled trial (*n* = 47)	Patients receiving *chemotherapy*		X	[80]
		0.1% chlorhexidine-based mouthwash	Double-blind, randomized controlled trial (*n* = 30)	Patients receiving *radiotherapy* for head and neck cancer		X	[81]
		Lozenges containing polymyxin E, tobramycin, and amphotericin B	Double-blind, randomized controlled trial (*n* = 65)	Patients receiving *radiotherapy* for head and neck cancer		X	[82]
		Oral paste containing polymyxin E, tobramycin, and amphotericin B	Double-blind, randomized controlled trial (*n* = 77)	Patients receiving *radiotherapy* for head and neck cancer		X	[83]
	Microbiota-targeted therapeutics	*Lactobacillus brevis* CD2 lozenges	Double-blind, randomized controlled trial (*n* = 31)	Patients receiving *chemoradiation* for head and neck cancer		X	[85]
		Capsule containing *Bifidobacterium longum*, *Lactobacillus lactis*, and *Enterococcus faecium*	Double-blind, randomized controlled trial (*n* = 99)	Patients receiving *chemoradiation* for head and neck cancer	✓		[84]
Preclinical		Oral microbiota transplantation (OMT)	Mice received OMT after treatment	Mice receiving head and neck irradiation	✓		[87]
	Novel innate immune-targeted therapeutics	TLR5 agonist CBLB502	Mice were injected subcutaneously with CBLB502 after treatment	Mouse model of head and neck cancer treated with radiotherapy	✓		[89]
		TLR5 agonist KMRC011	Beagle dogs were administered with KMRC011 up to 48 h after treatment	Beagle dogs receiving head and neck irradiation	✓		[90]
		Melatonin	Rats were administered with melatonin gel up to 21 days after treatment	Rats receiving tongue irradiation	✓		[92]
		Synthetic glycosaminoglycan GM-1111	Mice were administered with GM-1111 subcutaneously daily	Mice receiving head and neck irradiation	✓		[93]
		Immunomodulator peptide SCV-07	Golden Syrian hamsters were administered SCV-07	Hamsters receiving irradiation of buccal mucosa +/− chemotherapy	✓		[95]

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
