# Peer review of "The Role of the Innate Immune Response in Oral Mucositis Pathogenesis"

_ijms, 2023, doi:10.3390/ijms242216314_

Round 1

Reviewer 1 Report

Comments and Suggestions for Authors

Dear authors,

This review is well done. The etiopathogenesis of OM was very well deepened. There are some suggestions to improve the quality of this work

·      Please provide a material and methods section after the introduction, where you can explain how you conduct this review. Thanks.

·      Please improve the readability: there are too many long periods that are difficult to read by a not expert in the field. 

·      Please provide a summary table with all the therapies you suggest in preventing and treating OM in daily practice (to give a practical implication to theoretical concepts well analyzed and exposed).

Thank you.

Author Response

We thank reviewer 1 for their comments and suggestions. We have now created a table to summarise the interventions as requested. We expect the final edited version of the manuscript will have sufficient spacing for each paragraph to ensure readability.

Reviewer 2 Report

Comments and Suggestions for Authors

This review from Bowen and Cross summarizes the existing literature on the development of oral mucositis (OM) following cancer-related chemotherapy and radiation treatment and suggests several areas of focus for ongoing and future therapeutic development.  OM is a debilitating side effect of cancer treatment (particularly for head and neck cancers) with little available treatments outside of hygiene maintenance, pain management and generalized anti-inflammatories.  The authors focus on the activation of innate immunity pathways by cancer treatment-initiated tissue damage and the disturbance of homeostatic oral microbiome communities as under-explored contributors to OM, highlighting therapeutic options in these areas in pre-clinical or clinical stages of testing. 

As a whole, the review is well-written and discusses important burgeoning fields with respect to OM.  I have the following minor comments to be addressed  prior to acceptance for publication:

1) In addition to discussing the loss of tight junction proteins in section 3, the authors should also comment on the extent that gross tissue damage has been observed and quantified following cancer treatment that precedes OM development, in line with their model of OM outlined in figure 1.

2) In section 3.1, the authors should comment on whether significant differences has been observed for expression levels of TLRs at the oral epithelial surface versus the lamina propria and submucosa.

3) In section 3.2, the authors should comment on whether any specific bacterial species has been associated with the onset, prevention or resolution of OM.  The authors should also comment on whether cancer treatment has been shown to act as a bacterial selection event to expand resilient species.

4) Towards the end of section 3.2 where the authors discuss the limitations of germ-free mouse studies on OM development, the authors should briefly indicate the extent that adaptive (cellular or humoral) immunity has been associated with OM.

5) In figure 1, the disease stages (initiation, etc) shown at the bottom of the figure should be relocated to the top, since its location at the bottom associates it more with the layer of bone shown rather than the disease progression stages shown more at the top of the figure.  IL-17 should also be separated more from the inflammation and ulceration stage to distinguish it as a healing factor.

6) The authors should comment on whether IL-17 has been explored as a treatment option for OM, particularly since many of the treatment options discussed in section 4 appear to work most effectively at preventing OM when given co-current or following cancer treatment than at treating existing OM.

7) In section 4, the authors should clarify when discussing the positive impacts of digested probiotics on OM development is thought more to do with the medication being absorbed systemically and having comparable effects on the oral mucosa as well as the gut, or whether it's more to do with downstream effects of effecting the gut microbiome.

8) The therapeutics discussed in section 4 and 5 would benefit from being summarized in a table.
